# The Role of GSH in Intracellular Iron Trafficking

**DOI:** 10.3390/ijms22031278

**Published:** 2021-01-28

**Authors:** Robert Hider, Mayra Vera Aviles, Yu-Lin Chen, Gladys Oluyemisi Latunde-Dada

**Affiliations:** Institute of Pharmaceutical Science, King’s College London, London SE1 9NH, UK; mayra.vera_aviles@kcl.ac.uk (M.V.A.); yu_lin.chen@kcl.ac.uk (Y.-L.C.); yemisi.latunde-dada@kcl.ac.uk (G.O.L.-D.)

**Keywords:** cytosol, labile iron, glutathione, histidine, carnosine

## Abstract

Evidence is reviewed for the role of glutathione in providing a ligand for the cytosolic iron pool. The possibility of histidine and carnosine forming ternary complexes with iron(II)glutathione is discussed and the physiological significance of these interactions considered. The role of carnosine in muscle, brain, and kidney physiology is far from established and evidence is presented that the iron(II)-binding capability of carnosine relates to this role.

## 1. Introduction

Although iron transport is well characterised in bacteria [1], plants [2], and animals [3], the nature of the cytosolic and intracellular organelle iron pools remains uncharacterised. The so called “labile iron pool” first discussed by Greenberg and Wintrope [4] plays an essential role in supplying iron to the mitochondria for heme and iron–sulphur cluster synthesis, providing iron for the many cytosolic iron-dependent enzymes and, if in excess, presenting iron to ferritin and/or ferroportin. As free iron salts can catalyse the formation of toxic oxygen-containing radicals, the levels of this cytosolic pool must be tightly controlled; cells must be able to sense the iron levels and regulate iron homeostasis in order to maintain nontoxic levels of this key nutrient. Indeed, cytosolic iron(II) is believed to possess a control function by influencing the activity of a wide range of cytoplasmic-iron(II)-dependent enzymes [5], including the multitude of iron(II)-dependent 2-oxoglutarate oxygenases associated with histone and nucleic acid demethylation and protein hydroxylation [6].

The redox state of the cytosolic labile iron pool is iron(II) [5]. Williams argued that the electrode potential of the cytosol favours iron(II) over iron(III) and because the iron(II) binding constants for many cytoplasmic enzymes fall in the range 10^−8^–10^−7^ M [5,7], they require a similar standing concentration of iron(II) in order to prevent dissociation from the enzyme. Fluorescent probe studies, using calcein, has demonstrated that over 80% of the labile iron pool in K562 cells is iron(II) [8], a similar finding being reported for hepatocytes, using a fluorescent transition metal probe Phen Gren SK [9]. Thus, it is reasonable to conclude that the majority, if not the entire cytosolic labile iron pool is iron(II). In view of the 105-fold higher kinetic lability of iron(II) when compared with iron(III) [10], this is a logical strategy. Iron(II) is generally the form involved in intracellular translocation, for instance incorporation into iron-requiring enzymes, incorporation into ferritin, and transport across membranes by divalent metal transporter 1 (DMT1) and ferroportin.

Many ligands, previously linked with the cytosolic labile iron pool, will chelate iron(III) but not iron(II) at pH 7.0; namely, amino acids [11], ATP/AMP [12], inositol phosphates [13], and 2,5-dihydroxybenzoic acid [14]. In contrast, citrate does bind iron(II) under cytoplasmic conditions and has been suggested to be the major component of the labile iron pool [15]. A speciation plot ([iron(II)] = 1 μM; [citrate] = 100 μM), using previously determined affinity constants of citrate and iron(II) [16], confirms that iron(II) is predicted to be partially complexed by citrate at pH 7.0. However, Fe^II^·citrate is susceptible towards autoxidation at pH 7.0 [17], which renders citrate as an unlikely iron(II) buffer. Thus, it is probable that there is another ligand in the cytoplasm that is capable of coordinating iron(II).

## 2. Cytosolic Thiol-Containing Iron(II) Ligands

Based on the relatively strong interaction between iron(II) and thiol-containing compounds, H_2_S, cysteine, and glutathione (GSH) were considered as possible cytosolic ligands for iron(II) [18]. The cytosolic level of H_2_S is close to 15 nM [19,20] and using the stability constants for the Fe^II^-H_2_S interaction (log K_1_ = 5.1) [21], speciation plots demonstrate that at 100 nM and below, H_2_S cannot act as a cytosolic ligand for iron(II) [18]. Another potential ligand for iron(II) is cysteine (log K_1_ = 6.2) [22]. However, speciation plots based on the typical cytosolic levels of cysteine indicate that, as with H_2_S, this ligand does not bind iron(II) sufficiently tightly to make a significant contribution to the labile iron pool [18]. In contrast, GSH is present in the cytoplasm at a much higher concentration than either cysteine or H_2_S. The cytosolic level of GSH in human erythrocytes is 2.5 mM [23] and in rat liver, 8 mM [24].

Using the logK_1_ value for iron(II) of 5.12 [18], GSH is predicted to bind iron(II) to a considerable extent at pH 7.0, even at the lower extreme of cytosolic GSH concentrations, namely, 2 mM (Figure 1). Although cysteine, acting as a bidentate ligand (**2**, Scheme 1), possesses a higher log K_1_ value than GSH, the cytoplasmic GSH levels are two orders of magnitude higher than those of cysteine and thus GSH binds iron(II) more effectively in the cytosol. Furthermore, GSH rapidly reacts with hexaaquo·Fe^III^, converting it to hexaaquo·Fe^II^ (**1**) [25]. GSH does not chelate iron(II) as, unlike cysteine, the bidentate coordination would require the formation of rings containing either 9, 10, or 11 atoms. Such ring sizes are associated with unfavourable entropy differences. This non-chelating structure (**3**) (Scheme 1) agrees with that previously proposed for glutathione coordination to metals such as lead and mercury [26].

GSH dominates in competition with citrate at pH 7.0 (Figure 2). In the presence of 2 mM GSH, the iron(II) levels are effectively buffered by the large molar excess of the peptide. At equilibrium, with a total iron(II) cytosolic concentration of 1 μM (Figure 2), the concentrations of Fe^II^·GS (**3**), FeII·citrate, and hexaaquo·Fe^II^ (**1**) are, respectively, 8.2 × 10^−7^ M (82%), 3.1 × 10^−8^ M (3%), and 1.5 × 10^−7^ (15%). The percentage values remain relatively constant over the iron(II) range of 0.5–5 μM, and increasing the GSH concentration to 10 mM further favours the formation of Fe^II^·GS, increasing the percentage of iron(II) species to over 95%. Because of the high kinetic lability of iron(II), this system will be close to equilibrium and therefore at pH 7 the speciation plot depicted in Figure 2 will accurately reflect the cytosolic concentrations. Although autoxidation of cytosolic Fe^II^·GS will occur at a slow rate, the resulting iron(III) will be rapidly reduced back to iron(II) (Figure 3), in a fashion similar to that of the reduction of methemoglobin.

Iron(II) binding by GSH offers a means by which the cytosol can distinguish iron(II) and manganese(II), both of which are present at similar levels [27]. The affinity of manganese(II) for GSH is markedly lower than that of iron(II), the log K_1_ values being 2.7 [28] and 5.1 [18], respectively. This difference has a major effect on Mn(II) speciation, where citrate is found to be the dominant ligand, with GSH failing to bind even a trace of Mn(II) under cytoplasmic conditions [18].

Over the concentration ranges of GSH (2–10 mM) and iron(II) (0.5–5 μM) the two major cytosolic iron(II) species are reliably predicted to be Fe^II^·GS (**3**) and hexaaquo·Fe^II^ (**1**), with the glutathione complex dominating. It has been suggested that the 8000-fold excess of glutathione (8 mM) over coordinated iron(II) (1 μM) would minimise changes of [Fe^2+^] in the cytosol and render it unlikely that changes in the iron(II) concentration could act as a signal. However, the binding of glutathione to iron is pH dependent and at pH 7.0 approximately 95% of iron(II) is bound to glutathione, leaving the remaining iron as (Fe(H_2_O)_6_)^2+^, at a concentration of 4.5 × 10^−8^ M. The concentration of (Fe(H_2_O)_6_)^2+^ (**1**) is linearly related to the total concentration of cytosolic iron(II) and is predicted to vary between 10^−8^ M and 5 × 10^−7^ M for a range of total iron(II) concentration between 1 μM and 10 μM (Figure 4). Clearly, the large excess of glutathione does not prevent [Fe^2+^]_cytosol_ from acting as a potential signal for iron(II) sensors, such as the IRP1/aconitase- and IRP2/FBXL5-linked systems.

## 3. Mitochondrial Labile Iron Pool and Iron Sulphur Cluster Biosynthesis

Mitochondria are major sites for heme and iron–sulphur cluster synthesis in both plants and animals and consequently there is a constant iron influx into mitochondria, as they export both heme and iron–sulphur clusters for use elsewhere in the cell. There are almost certainly multiple mechanisms for mitochondrial iron uptake. Mitoferrins 1 and 2 have, for instance, been linked with the delivery of iron to mitochondria in a range of organisms [29,30], the accumulation being dependent on the mitochondrial membrane potential [31]. Significantly, there is also a considerable movement of glutathione across mitochondrial membranes, facilitated by the dianionic exchangers, namely, malate^2−^/HPO_4_^2−^ and 2-oxoglutarate^2−^/malate^2−^ [32,33]. An interesting possible variant of such an exchange is for Fe^II^GS (**3**) to act as a substrate for this family of transporters. Indeed, it has been suggested that iron(II) is transported into the mitochondrion as a complex rather than an inorganic iron [31].

The concentrations of citrate and glutathione are higher in the mitochondrial matrix than in the cytosol; namely, [citrate] = 1.1–1.25 mM [34,35] and [GSH] = 11 mM [24]. Speciation analysis confirms that, under these conditions, Fe^II^GS is the dominant form of iron(II) at pH 8.0 (data not shown). Indeed, a pH of 8.0 further favours the formation of Fe^II^GS (**1**) when compared to the cytosol. Furthermore, the mitochondrial ratio of [GSH]/[GSSG] is higher than that found in the cytosol [36]. This mitochondrial labile iron pool, Fe^II^GS, will also supply iron to mitochondrial ferritin [37].

In addition to an iron(II) buffering role, Fe^II^GS binds to glutaredoxins, proteins which are required for iron cluster assembly and heme biosynthesis [38]. Glutaredoxins are widely distributed; being found in virus particles, bacteria, yeast, plants, and mammals [39]. They are small proteins that possess a glutathione binding site and a redox active thiol function, both of which are located on the surface of the protein. These thiol functions typically catalyse thiol-disulphide exchange reactions. Some glutaredoxins are also capable of simultaneously binding iron and glutathione, forming [2Fe-2S] clusters that are shared between two subunits of a homodimer [39,40]. These [2Fe-2S] clusters are transferred to acceptor proteins [41,42,43] and are involved in iron-response protein (IRP) regulation [44]. The mitochondrion is the dominant location for iron-sulphur cluster synthesis [45], but the precise steps in [4Fe-4S] cluster biosynthesis are not completely defined [46], although glutathione has been demonstrated to possess a central role in the process [47,48]. The overall biogenesis occurs in two parts; the de novo assembly of an Fe–S cluster on a scaffold protein, such as the IscU, and then the subsequent transfer to target the apoproteins. Glutaredoxins and glutathione are directly involved in these processes [39,49,50]. In the presence of glutathione, [2Fe-2S] clusters undergo a reversible exchange between apo ISU and glutathione, forming a complex [51]. Reductive coupling of two [2Fe-2S] clusters to form a single [4Fe-4S] cluster takes place on homodimeric cluster scaffold proteins [50] and the resulting cluster can, in turn, be transferred to apoproteins. Thus, it is conceivable that glutathione acts not only as a ligand for mononuclear iron(II) but also for [2Fe-2S] clusters.

## 4. Intracellular Distribution of Iron

As well as presenting iron to cytosolic apo-iron-dependent enzymes [6] and iron-sensing proteins, such as IRP1/aconitase and IRP2/FBXL5 [52], the labile iron pool has to be able to direct iron to ferritin and ferroportin when the concentration of the labile iron pool begins to approach a critical elevated level. How can this be achieved?

It has been suggested that the labile iron pool is directed to some intracellular sites via protein chaperones [53,54], but a related possibility is direction via low molecular weight ternary glutathione complexes. Both mechanisms will be considered below.

Iron chaperones: Cytosolic iron(II)-containing enzymes and cytosolic iron(II) sensors possess metal-binding sites that bind iron(II) in preference to zinc(II) and copper(II) under physiological conditions. This is achieved by maintaining the cytosolic concentrations of zinc(II) and copper(II) at a much lower level than that of iron(II), namely, 10^−11^ M and 10^−15^ M, respectively, as compared to iron(II) at 10^−6^ M [55,56]. Iron(II) is capable of binding rapidly to nascent apoproteins, the iron(II) levels being sufficient for these sites to be largely occupied. As there are several hundred such iron(II)-dependent enzymes in the cytosol, it seems unlikely that chaperones are required to supply iron to all these enzymes. However, the direction of iron to ferritin, ferroportin, and the mitochondria could well require a chaperone type mechanism.

Poly r C-binding protein 1 (PCBP1) binds iron and has been reported to deliver iron to specific proteins, for instance ferritin [53] and the cytosolic [2Fe-2S] cluster assembly [54]. In the latter interaction, iron is presented to PCBP1 as iron(II)glutathione, forming a ternary complex. This complex then interacts with BolA2 and the iron is subsequently processed. In the absence of GSH this process does not occur [54]. A similar mechanism possibly operates for the PCBP1-mediated donation of iron to ferritin. However, PCBP iron chaperones are apparently not required for delivery of iron to mitochondria [57].

Low molecular weight iron–glutathione ternary complexes: Iron(II)glutathione, under physiological conditions ([Fe] = 1 μM; [GSH] > 2 mM; pH 7.0), exists as a hydrated iron(II) cation with a single coordination site being occupied by a monodentate glutathione molecule (**3**). In principle, other low molecular weight ligands present in the cytosol can simultaneously coordinate to this ion, forming a ternary complex (**4**) (Scheme 2). The additional bidentate or tridentate ligand would need to possess a similar affinity for iron(II) as that of GSH in order to form an appreciable concentration of the ternary complex; the ligand would also need to be present at a relatively high concentration (1–10 mM). These requirements severely limit the number of such possibilities. Histidine emerges as one possible candidate, it having an appreciable affinity for iron(II) (logK = 5.85, cf logK for GSH = 5.1), and is universally present in mammalian cells (150–600 μM). Significantly, histidine has a protective property against oxidative stress damage in the kidney and this may result, at least partially, from its iron-coordinating properties [58]. The speciation plot of iron(II) and histidine is presented in Figure 5A; it is clear that histidine is capable of binding iron(II) at pH 7.0, with approximately 70% of the iron(II) being coordinated by histidine; but, in the presence of glutathione (2 mM), this percentage is much reduced. (Figure 5B). However, when the speciation analysis of the cytosolic fluid includes the ternary complex iron(II)·glutathione·histidine (**5**) (Scheme 2) (Figure 5C), the amount of iron binding to the ternary complex is found to be equal to that of iron(II)·glutathione. Clearly, the iron(II)·glutathione·histidine complex (**5**) (Scheme 2) can form under normal physiological conditions. There are many differences between Complexes **3** and **5**, including the net charge, hydrophilicity, and hydrogen bond formation potential. These differences could be utilized by the cytosol to direct iron to different targets.

Significantly, a group of widely distributed histidine-containing dipeptides, including carnosine (Scheme 3), also possess the ability to provide protection against oxidative stress [59,60]. Carnosine (**6**) is a well-established chelating molecule, capable of binding transition metals [61]. Indeed, it has been suggested that carnosine can compete for iron(II) with hypoxia-inducible factor 1 alpha (H1F-1α), an iron-dependent proline hydroxylase [62]. Furthermore, carnosine has previously been demonstrated to form mixed transition metal complexes with glutathione [63]. A likely structure for such a ternary complex of carnosine, glutathione and iron(II) is indicated in Structure **7** (Scheme 2).

## 5. Carnosine and Histidine—Iron(II) Chelators of Physiological Significance

Carnosine and its closely related analogues are found at relatively high levels in skeletal muscle (2–20 mM) and cardiac muscle (2–10 mM), with lower levels found in the brain, liver, and kidney. These histidine-containing dipeptides are reported to possess antioxidant properties and can inhibit glycoxidation and protein carbonylation [64]. Such properties have been associated with the metal-complexing properties of the histidine-containing peptides and also the free radical quenching properties of the imidazole ring [65]. Most studies have been centred on the chelation of copper(II) and zinc(II) [59,61]. However, because of the extremely low concentrations of these two cations in the cytosol, it is unlikely that their ability to form carnosine or histidine complexes is physiologically relevant [56]. The situation with iron(II) could be different due to the much higher cytosolic concentration (1–5 μM) [56]. Surprisingly, there are few reports centred on the coordination of iron(II) by the histidine-containing dipeptides. This is undoubtedly due to the tendency of ferrous ion salts to form insoluble polymeric complexes with carnosine and related dipeptides in aqueous solution at concentrations typically adopted for spectroscopic measurement. Consequently, whereas affinity constants of carnosine have been reported for Ca^2+^, Mg^2+^, Cd^2+^, Mn^2+^, Ni^2+^, Cu^2+^, and Zn^2+^, no such value has been reported for Fe^2+^ [61]. However, by considering the trend of these affinity constants in the Irving–Williams series, it is possible to reliably estimate a value for the affinity constant of iron(II), namely, logK_1_(Fe^2+^) = 5.1. This is slightly lower than the corresponding value for histidine (logK_1_(Fe^2+^) = 5.85). Using this value it is possible to study the speciation plots for iron(II) carnosine in the presence and absence of glutathione. In the absence of glutathione, iron(II) is coordinated by carnosine at physiological levels (Figure 6A), appreciable chelation occurring at pH values of 7 and above. When glutathione (2 mM) is present together with carnosine (10 mM), as with histidine, competition occurs between the two compounds for iron(II), glutathione dominating at pH values above 7.0 (Figure 6B). This simulation presented in Figure 6B assumes that no ternary complex is formed. When the ternary complex glutathione-Fe(II)-carnosine (**7**) (Scheme 2) is included in the speciation study, the ternary complex is predicted to dominate over the pH range 8–10 and the concentration ratio of Complexes **3** and **7** is approximately 1:1 at pH 7.0 (Figure 6C). Clearly, as with histidine, the presence of carnosine will influence the cytosolic level of iron(II)glutathione and will lead to the formation of an additional iron complex (**7**) in the cytosol (Scheme 4). As with **5**, Complex **7** could in principle direct iron to different protein targets to those targeted by iron(II)glutathione (**3**).

Surprisingly, the physiological role of the histidine-containing dipeptides, including carnosine, is unclear [59]. There is no doubt that the presence of carnosine is important for muscle contraction, including that of cardiac muscle [59]. Recently, it has been suggested that carnosine protects muscle against post ischemia by augmenting HIF-1α angiogenic signalling by iron chelation [62]. Significantly, methylcarcinine (Scheme 3), a carnosine analogue, lacking iron(II)-chelating capacity, was found to have no effect on HIF-1α activity or the release of vascular endothelial growth factor in complete contrast to carnosine [62]. Thus, the iron-chelating properties of carnosine appear to be relevant to its role in muscle.

Likewise, a protective role for carnosine has also been reported in diabetes [66,67]. Indeed, carnosine concentrations in the livers of diabetic mice are lower than those in control mice [68]. In this same study, carnosine was reported to decrease the circulating insulin-like growth factor-binding protein (IGFBP) levels, through a mechanism that involved the suppression of HIF-1α-mediated IGFBP induction [68]. Again HIF-1α, an iron(II)-dependent enzyme, appears to be involved with the mode of action of carnosine. Carnosine can be localised in different compartments of tissues, for instance the kidney, by controlling the location of the enzymes involved in the synthesis and the breakdown of the dipeptide (Scheme 3) [69]. Carnosine can also be enzymatically converted to a range of analogues (Scheme 3), some of which chelate iron and others that do not. Clearly, there are multiple means of controlling the histidine-containing dipeptides levels.

GSH is an antioxidant tripeptide that serves as a cofactor for glutathione peroxidase 4 (GPX4), the sole selenoenzyme that catabolizes the reduction of phospholipid hydroperoxides. It is therefore important to maintain high glutathione levels (2–10 mM) in order to protect membrane lipids. Increased oxidative stress in the mitochondria and/or the cytosol, which is linked to iron accumulation, can lead to GSH depletion and GPX4 inactivation, culminating in the initiation of degenerative diseases, such as Alzheimer’s disease, Parkinson’s disease, amyotrophic lateral sclerosis, and ischemia/reperfusion injury [70,71]. The depletion and inhibition of GSH antioxidant levels inactivate and repress the decomposition of lipid peroxides into lipid alcohols. Orchestration of this metabolic derangement has been described as a regulated cell-death process termed “ferroptosis” [72]. Interestingly, carnosine and histidine have been associated with the increased expression of catalase and glutathione peroxidase, two key antioxidant enzymes [73,74].

The roles of glutathione, carnosine, and histidine in the cytosolic labile iron pool need to be thoroughly investigated. Furthermore, studies aimed at elucidating the candidacy of the Fe^II^GS complex for the in situ mitochondrial labile pool and as the putative iron complex that is transported into the mitochondria are imperative.

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
