# Peer review of "The Role of GSH in Intracellular Iron Trafficking"

_ijms, 2021, doi:10.3390/ijms22031278_

Round 1
Reviewer 1 Report
The authors well described intracellular iron ion complexes and GSH function in different cell organelles.
I would like to suggest provide detailed information of some abbreviation, such as DMT1, IRP.
Author Response
Reviewer 1 was complimentary to our submission but suggested that we provide a list of abbreviations. We have introduced such a list before the reference section.
Reviewer 2 Report
This is a very interesting review considering the role of glutathione as a ligand for cytosolic iron. I would recommend adding a section where the importance of glutathione as an iron ligand with respect to specific diseases, such as neurodegenerative diseases, would be discussed. This would support the importance of the issue discussed for the readers.
Author Response
Reviewer 2 finds that our manuscript is interesting, but suggested that we should add a section discussing the relationship of glutathione with neurodegeneration. This is a good suggestion and we have introduced a new paragraph discussing this matter. This is the penultimate paragraph in section 5. We have introduced four additional references in this paragraph.